# Advances and Prospects of Virus-Resistant Breeding in Tomatoes

**DOI:** 10.3390/ijms242015448

**Published:** 2023-10-22

**Authors:** Zolfaghar Shahriari, Xiaoxia Su, Kuanyu Zheng, Zhongkai Zhang

**Affiliations:** 1Biotechnology and Germplasm Resources Research Institute, Yunnan Academy of Agricultural Sciences, Yunnan Seed Laboratory, 2238# Beijing Rd, Panlong District, Kunming 650205, China; shahriari225@gmail.com (Z.S.); sxx919@163.com (X.S.); 2Crop and Horticultural Science Research Department, Fars Agricultural and Natural Resources Research and Education Center, Agricultural Research, Education and Extension Organization (AREEO), Shiraz 617-71555, Iran

**Keywords:** virus resistance gene, RNAi, tomato breeding, CRISPR-Cas, genome editing

## Abstract

Plant viruses are the main pathogens which cause significant quality and yield losses in tomato crops. The important viruses that infect tomatoes worldwide belong to five genera: *Begomovirus*, *Orthotospovirus*, *Tobamovirus*, *Potyvirus*, and *Crinivirus*. Tomato resistance genes against viruses, including *Ty* gene resistance against begomoviruses, *Sw* gene resistance against orthotospoviruses, *Tm* gene resistance against tobamoviruses, and *Pot 1* gene resistance against potyviruses, have been identified from wild germplasm and introduced into cultivated cultivars via hybrid breeding. However, these resistance genes mainly exhibit qualitative resistance mediated by single genes, which cannot protect against virus mutations, recombination, mixed-infection, or emerging viruses, thus posing a great challenge to tomato antiviral breeding. Based on the epidemic characteristics of tomato viruses, we propose that future studies on tomato virus resistance breeding should focus on rapidly, safely, and efficiently creating broad-spectrum germplasm materials resistant to multiple viruses. Accordingly, we summarized and analyzed the advantages and characteristics of the three tomato antiviral breeding strategies, including marker-assisted selection (MAS)-based hybrid breeding, RNA interference (RNAi)-based transgenic breeding, and CRISPR/Cas-based gene editing. Finally, we highlighted the challenges and provided suggestions for improving tomato antiviral breeding in the future using the three breeding strategies.

## 1. Introduction

Tomato (*Solanum lycopersicum* L.) is one of the most economically valuable fruit or vegetable crops worldwide. According to the Food and Agriculture Organization of the United Nations (FAO) Statistical report, the total worldwide production of tomatoes was 189.23 million tons in 2021, with a value of over USD 30 billion. Viral diseases can significantly decrease the yield and quality of tomatoes [1]. According to the new classification system (2022) approved by the International Committee for the Classification of Viruses (ICTV), there are 181 viral species infecting tomato crops. The major tomato viral pathogens that have been emerging worldwide over the past 20 years include the following genera: *Begomovirus*, *Orthotospovirus*, *Tobamovirus*, *Potyvirus*, and *Crinivirus* [1,2,3].

Considering that most of these viruses are transmitted by insect vectors, the main agronomic and classic management of viral diseases involves controlling the vector with insecticides and uprooting symptomatic plants or those with sanitary voids to reduce the incidence of the virus [4,5]. However, these controlling methods only reduce the viral effects to some extent and cannot efficiently eliminate tomato virus disease [6]. Virus resistance breeding is the most promising method for controlling viral diseases [7,8]. Combining conventional hybrid breeding with marker-assisted selection (MAS) to introduce resistance genes from wild germplasm into cultivated cultivars has proven to be effective for virus resistance breeding in tomatoes [8,9,10]. However, most of the virus resistance genes currently found in tomatoes are single-gene-mediated with qualitative resistance, and thus, virus mutation and mixed infection can easily lead to resistance breakdown [11,12,13]. In addition, some emerging tomato viruses, such as tomato chlorosis virus (ToCV), tomato brown rugose fruit virus (ToBRFV), and tomato mottle mosaic virus (ToMMV), still lack relevant natural resistance genes, making conventional virus resistance breeding in tomatoes challenging [14,15,16].

This review summarizes tomato viruses and their characteristics. According to the epidemic characteristics of tomato viruses, we propose that tomato virus resistance breeding should focus on rapidly, safely, and efficiently creating broad-spectrum germplasm materials resistant to multiple viruses. Based on this proposition, we summarize and analyze the advantages and characteristics of three tomato antiviral breeding strategies: MAS-based hybrid breeding, RNA interference (RNAi)-based transgenic breeding, and CRISPR/Cas-based gene editing. Finally, we discuss the challenges and provide suggestions for improving these three breeding strategies in the future.

## 2. Tomato Viruses and Their Epidemic Characteristics

Currently, the main tomato epidemic viruses worldwide include begomoviruses, orthotospoviruses, tobamoviruses, potyviruses, and criniviruses. The classification and details of these viruses are listed in Table 1. These major epidemic viruses generally cause tomato plant leaf shrinkage, chlorosis, leaf and fruit necrosis, spotting, and other symptoms, which significantly impact tomato quality and yield (Figure 1).

A typical characteristic of plant viruses is the rapid nucleotide mutation and genome recombination rates, which contribute to the emergence of novel viral strains or species, increase the virulence of the virus, and cause the breakdown of host resistance, resulting in severe symptoms in the host plant [17,18,19].

Another characteristic of plant viruses, as revealed by high-throughput sequencing or other means of detection, is the presence of high mixed-infection incidence in the field between viruses or between viruses and other pathogens [2,20,21]. Mixed infections usually increase synergies between pathogens and the breakdown of host resistance, which has been reported in many cases [13,22]. Therefore, developing a quick response against the resistance breakdown caused by virus mutation, recombination, and mixed infection to efficiently create broad-spectrum and persistent antiviral germplasm materials has become a major challenge and the primary objective in tomato antiviral breeding.

**Table 1 ijms-24-15448-t001:** Classification and details of the major epidemic viruses of tomato.

Genus	Epidemic Species Worldwide	Genome	Transmission	Symptoms	References
*Begomovirus*	Tomato yellow leaf curl virus (TYLCV)) and tomato leaf curl virus (ToLCV)	Single-stranded DNA (ssDNA)	Whitefly, seed	Yellowing, curling, and a significant loss in apical leaf. Early-infected plants are frequently infertile. Since most blooms (>90%) droop after infection, there is almost no or fewer small fruit.	[23,24,25,26]
*Orthotospovirus*	Tomato spotted wilt virus (TSWV)	Negative-sense single-stranded ambisense (-ssRNA) RNA	Thrips, seed	Stunting, necrosis, bronzing, chlorosis, ring spots, and ring patterns on the leaves, stems, and fruits.	[27,28,29]
*Tobamovirus*	Tobacco mosaic virus (TMV), tomato mosaic virus (ToMV), and tomato brown rugose fruit virus (ToBRFV)	Single-stranded positive-sense RNA (+ssRNA)	Seed, mechanical transmission such as by hand, pruning tools, soil, etc.	Yellow–green mottling on the leaves; stunted growth; flowers and leaflets may be curled, distorted, and smaller than normal in size.	[15,30,31,32,33,34]
*Potyvirus*	Potato virus Y (PVY), and, chilli veinal mottle virus (ChiVMV)	Single-stranded positive-sense RNA (+ssRNA) viruses	Aphid, seed	Leaf mosaic, mottle and crinkling, vein necrosis and necrotic spots, stem and petiole necrosis, leaf drop, and yield reduction.	[2,21,35,36]
*Crinivirus*	Tomato chlorosis virus (ToCV)	Single-stranded positive-sense RNA (+ssRNA) viruses	Whitefly	Leaf chlorosis, chlorotic flecking, and bronzing. Fruits are symptomless but with reduced yield.	[37,38]

## 3. Strategies of Tomato Resistance Breeding against Viruses

At present, there are three main strategies for tomato antiviral breeding: MAS-based hybrid breeding, RNAi-based transgenic breeding, and CRISPR/Cas-based gene editing. (1) MAS-based hybrid breeding. In this strategy, wild or domestic germplasm with resistance genes is hybridized with non-resistance germplasm materials, and F1 hybrid generation is further selfed for multiple generations. By serving as a resistance selection method for the selfed generations, MAS allows for the rapid creation of new resistance germplasm (Figure 2A) [39]. (2) RNAi-based transgenic breeding. In this strategy, a targeted virus gene/dsRNA/microRNA sequence is transferred into the non-resistant germplasm material by the *Agrobacterium*-mediated transformation method so as to induce the RNA silencing effect in the host plant to resist virus infection (Figure 2B) [40]. (3) CRISPR/Cas-based gene editing. In this strategy, a CRISPR/Cas vector is designed to target the DNA/RNA sequence of the virus or the host’s susceptible genes, thus interrupting the replication and assembly process of the virus in the host, ensuring resistance against the virus (Figure 2C) [41]. These three strategies are discussed in the following sections.

## 4. Virus Resistance Gene and MAS-Based Hybrid Breeding in Tomatoes

### 4.1. The Tomato Ty Gene Family Encoding for Resistance against Begomoviruses

At present, six begomoviruses resistance genes, all belonging to the *Ty* gene family, have been identified, namely, *Ty-1*, *Ty-2*, *Ty-3*, *Ty-4*, *Ty-5*, and *Ty-6* (Table 2). *Ty-1*, *Ty-3*, *Ty-4*, and *Ty-6* are derived from the wild tomato species *S. chilense,* while *Ty-2* and *Ty-5* are obtained from *S. habrochaites* and the commercial tomato cultivar *Tyking*, respectively [42,43]. *Ty-1* is allelic with *Ty-3*, and the two genes are located on chromosome 6 of *S. chilense* [44]. *Ty-1*/*3* increases the cytosine methylation of viral genomes and induces a hypersensitive response to viral infections, conferring plants with TYLCV resistance [45]. Although *Ty-1* exhibits broad-spectrum begomoviruses resistance, recent studies have shown that resistance is compromised by the co-infection with a beta satellite [46]. Gene *Ty-2* located on the long arm of *S. habrochaites* chromosome 11 has been identified as the nucleotide-binding domain and a leucine-rich repeat-containing (NB-LRR) gene [47]. Moreover, the *Ty-4* gene maps to chromosome 3 of *S. chilense* and has been reported to increase virus resistance in combination with *Ty-3* [42]. The recessive TYLCV resistance gene *Ty-5* located on chromosome 4 of the commercial tomato cultivar *Tyking* encodes the mRNA surveillance factor *Pelota* [8,42]. A recent study showed that *Ty-5* confers broad-spectrum resistance to two representative begomoviruses occurring in China [8]. *Ty-6*, located on chromosome 10 of *S. chilense*, effectively complements the resistance conferred by *Ty-3* and *Ty-5* [48]. It was reported that *Ty-6* also confers resistance to tomato mottle virus (ToMoV), suggesting that the gene inhibits both mono- and bi-partite begomoviruses in tomatoes [48]. Although PCR-based markers have been identified and developed for *Ty-1*, *Ty-2*, *Ty-3*, and *Ty-4* TYLCV-resistant loci, these markers are inconsistent, thus limiting their application in MAS [49,50].

### 4.2. The Tomato Sw Gene Family Conferring Resistance against Orthotospoviruses

*S. Peruvianum*, reported as the first wild tomato with a broad-spectrum resistance to tomato spotted wilt virus (TSWV), has been widely crossed with commercial cultivars since the 1930s [7]. Studies on molecular genetics indicated that this resistance was conferred by a single dominant gene/locus named *Sw-5*, which was initially found to be effective against several TSWV isolates from the United States [64] and Brazil [65]. *Sw-5* also exhibits a broad spectrum and high level of resistance to other orthotospoviruses, including tomato chlorotic spot virus (TCSV), chrysanthemum stem necrosis virus (CSNV), and groundnut ringspot virus (GRSV), and is widely used in tomato breeding [7,66]. *Sw-5* was discovered near the telomeric area of chromosome 9 between the CT71 and CT220 restriction fragment length polymorphism (RFLP) markers [67]. The *Sw-5* locus is part of a loosely clustered gene family containing six paralogous genes: *Sw-5a, Sw-5b, Sw-5c, Sw-5d, Sw-5e,* and *Sw-5f* [7,10]. Among these genes, only *Sw-5b* has universal resistance to various TSWV isolates, although *Sw-5a and Sw-5b* are highly homologous (95%) [10,68]. In addition to the *Sw-5* gene, seven other genes resistant to TSWV belonging to the *Sw* gene family have been currently identified in tomatoes, and they include *Sw-1a*, *Sw-1b*, *Sw-2*, *Sw-3*, *Sw-4*, *Sw-6,* and *Sw-7* (Table 2). The introgression/incorporation of these resistance alleles in the commercial varieties could create materials with broad-spectrum resistance [69]. Molecular markers associated with TSWV resistance in tomatoes were summarized in a previous study that developed more than 20 molecular linkage markers for *Sw-5* and *Sw-7* TSWV resistance genes [10]. Some of these linkage markers included randomly amplified polymorphic DNA (RAPD), a sequence-characterized amplified region (SCAR), amplified fragment length polymorphisms (AFLP), cleaved amplified polymorphic sequence (CAPS), insertion–deletion (In-DEL), SNP, competitive allele-specific PCR (KASP), RFLP, and simple sequence repeats (SSR). However, the linkage molecular markers of *Sw-1a*, *Sw-1b*, *Sw-2*, *Sw-3*, *Sw-4*, and *Sw-6* have not been reported yet [10].

### 4.3. The Tomato Tm Gene Family Conferring Resistance against Tobamoviruses

For more than five decades, three resistance genes, *Tm-1*, *Tm-2*, and *Tm-2^2^*, have been considered as the resistance factors against tobamoviruses in tomatoes (Table 2) [55,56,70]. *Tm-1* is an incompletely dominant gene derived from the wild tomato *S. habrochaites* [14,55]. Conversely, *Tm-2* and *Tm2^2^* are completely dominant genes introgressed from *S. peruvianum* and are considered allelic [14]. *Tm-2^2^* confers more effective resistance than *Tm-1* or *Tm-2* and has shown durable ToMV resistance for the last 60 years, explaining why it is the most currently and widely utilized in breeding tomato cultivars [71]. Reports showed that ToBRFV overcomes all the tobamoviruses resistance genes in tomatoes, including the durable *Tm-2^2^* resistance gene [57]. The PCR-based markers for *Tm-1*, *Tm-2*, and *Tm-2^2^* resistant genes have also been reportedly used for MAS [50,72,73]. Despite the many advantages and several reports on its application in studying virus resistance genes, MAS is still in the research stage of its utilization in breeding tomatoes for virus resistance [50].

### 4.4. The Tomato Pot 1 Gene Conferring Resistance against Potyviruses

The wild tomato relative *Lycopersicon hirsutum* PI247087 was identified as the source of resistance to potyviruses [61,62]. Analysis indicated that resistance is conferred by a single recessive gene, *pot-1*, mapped to the short arm of tomato chromosome 3 in the vicinity of the recessive *py-1* locus for resistance to corky root rot [61]. Studies revealed that the recessive *pot-1* gene is the orthologue of the pepper (*Capsicum annuum*) *pvr2* gene [63].

### 4.5. The Challenges in Breeding Tomato Hybrid Cultivars against Viruses

Although using natural virus-resistant germplasm resources for hybrid breeding is the preferred and most effective way for obtaining resistant commercial varieties, only a few virus-resistant tomato germplasm resources are available, and the resistance is limited to a few viruses [74]. For example, virus resistance hybrid breeding cannot be applied against emerging viruses, such as ToCV, ToBRFV, ToMMV, etc., due to the lack of their corresponding natural resistance genes [14,15,16].

The other challenge is that most of the virus-resistance genes discovered so far exhibit single-gene-mediated qualitative resistance, which can be easily broken by virus mutations and mixed-infection, a phenomenon that has been commonly reported [12,13,75]. Improving multiple virus-resistant lines and cultivars by introducing several resistance genes targeting different viruses in a cultivar may play an important role in future tomato improvement projects [76,77,78,79]. Though promising, this process is complex and requires very long cycles of hybrid breeding. One example is a homozygous breeding line UMH 1203 carrying the *Tm-2^a^*, *Ty-1*, and *Sw-5* genes, which took 10 years to successfully develop multiple resistance against ToMV, TSWV, and TYLCV from several tomato landraces [78,80]. Nevertheless, a yield reduction of 40–50% has been reported for this breeding line at low-virus-incidence conditions [80]. Several reports indicated that the introgression of TYLCV resistance caused most of the yield reduction observed in fresh tomatoes due to the introgressed genes and/or linkage drag from the wild tomato species [80,81]. This reduction in agricultural yield would only be acceptable when cultivating under high levels of virus infection, and these new multi-resistance lines should only be used to develop cultivars for highly virus-infected areas [81].

## 5. RNAi-Based Transgenic Breeding

RNA silencing, also known as RNAi, is a conserved defense mechanism that suppresses the expression of viral nucleic acids, transposable elements, or host genes that need to be regulated [40,82]. The principle is based on the recognition and splicing of the double-stranded (ds) or hairpin (hp) RNA by Dicer-like (DCL) proteins into 21- to 24-nucleotide (nt) small RNAs (sRNAs). The subsequent steps involve a series of sRNA signal amplification and cleavage target mRNA processes with the participation of RNA-induced silencing complex (RISC) [83]. RNAi is an important pathway for plants resisting viral infections through the sequence-specific degradation of target viral RNA [40,74,84,85,86].

In the first-generation antiviral transgenic strategy, a single-stranded sequence of a viral gene, such as viral coat protein (CP) gene, RNA dependent RNA polymerase (RdRp) gene, etc., is transferred into the host plant genome, inducing the RNA silencing effect against the target virus by the host (Table 3). Subsequent studies found that transferring double-stranded RNA (dsRNA) or hairpin RNA (hpRNA) constructed based on viral sequences was more effective in inducing the RNA silencing effect than single-stranded viral sequences, making it a second-generation antiviral transgenic strategy (Table 3). With the recent depth of research on sRNAs, a new antiviral gene transfer strategy based on artificial sRNA engineering technology has been developed [87]. This third-generation antiviral transgenic strategy is based on artificial microRNAs (amiRNAs) or synthetic trans-acting small interfering RNAs (syn-tasiRNAs), which are 21 nt and artificially engineered to be highly specific to ensure a high sequence complementarity with target virus RNA and overcome the limited specificity of RNAi [88,89]. AmiRNAs and syn-tasiRNAs are functionally similar but are generated differently. AmiRNAs are derived from the DCL1 cleavage of miRNA precursors with foldback structures, while syn-tasiRNAs are produced in a multi-step RNAi process [90].

The biggest advantage of the third-generation antiviral transgenic strategy based on amiRNAs and syn-tasiRNAs is that it aggregates multiple-virus-targeting, which rapidly creates broad-spectrum resistance [91,92,93]. Many successful cases have been reported in this regard. For example, *Arabidopsis* miR159 was used as a backbone to express genes targeting P25, HC-Pro, and Brp1 of potato virus X (PVX), potato virus Y (PVY), and potato spindle tuber viroid (PSTVd) via the third-generation antiviral transgenic strategy, demonstrating resistance against PVX, PVY, and PSTVd co-infection simultaneously [92]. Another study showed that the *Arabidopsis TAS3a* gene was engineered to express syn-tasiRNAs targeting the genome of turnip mosaic virus (TuMV) and cucumber mosaic virus (CMV). The transgenic *Arabidopsis thaliana* plants expressing these syn-tasiRNAs showed high levels of resistance to both viruses [91].

In general, the third-generation antiviral transgenic strategy has shown high potential for virus resistance breeding. However, public concern and controversy over genetically modified (GM) crops and the strict regulation of policies have greatly inhibited the commercial development potential of GM crops. Therefore, some researchers opt to apply amiRNAs or dsRNA exogenously as crude extracts to prevent virus infection, which can also effectively induce the gene-silencing pathway in host plants against virus infection [94,95].

**Table 3 ijms-24-15448-t003:** RNAi-based transgenic virus breeding methods in plants.

Strategy	Target Virus	Genus	RNAi Induction Method	Targeted Region	Precursor(s)	Efficiency	Reference
First-generation antiviral transgenic strategy	TMV	*Tobamovirus*	ssRNA	*CP*	cDNA	Delayed symptom development; 10 to 60 percent of the transgenic plants failed to develop symptoms.	[96]
	TMV	*Tobamovirus*	ssRNA	*CP*	cDNA	The resistance level of expression TMV CP from the pal2 promoter is less than that of the 35S promoter.	[97]
	TSWV	*Orthotospovirus*	ssRNA	*N*	cDNA	Lack of systemic symptoms and little or no systemic accumulation of virus.	[98]
	ToLCV	*Begomovirus*	ssRNA	*Rep*	cDNA	A high level of resistance and inheritability of the transgene was observed up to T2.	[99]
	TLCV	*Begomovirus*	ssRNA	*CP*	cDNA	T1-generation transgenic plants were showed variable degrees of disease resistance/tolerance compared to the untransformed control.	[100]
	ToLCNDV	*Begomovirus*	ssRNA	*AV2*	cDNA	Transgenic plants showed symptomless, although viral DNA could be detected in some plants by PCR.	[101]
	PRSV	*Potyvirus*	ssRNA	*CP*	cDNA	PRSV infection was not observed on any of the transgenic resistance (TR) plants. TR plant yields were at least three times higher than the industry average.	[102]
Second-generation antiviral transgenic strategy	TMV	*Tobamovirus*	dsRNA	*CP*, *p126*	dsRNA	The application of TMV p126 dsRNA onto tobacco plants induced greater resistance against TMV infection as compared to CP dsRNA (65 vs. 50%).	[103]
	ToLCV	*Begomovirus*	hpRNA	*AC1*, *AC4*	hpRNA	Provides a promising approach to suppress a wide spectrum of ToLCV infection in the tomato.	[104]
	ToLCV	*Begomovirus*	dsRNA	*AC4*	dsRNA	Absolute absence of leaf curl virus disease symptoms and reduction in nematode symptoms.	[105]
	ToCMoV	*Begomovirus*	hpRNA	*AC1*, *AC4*, *AV1*, *AC5*	hpRNA	Most transgenic lines showed significant delays in symptom development, and two lines had immune plants.	[106]
	PVY	*Potyvirus*	dsRNA	*CP*	dsRNA	Highly resistant to three strains of PVY.	[107]
	PVY	*Potyvirus*	hpRNA	*CP*	hpRNA	Nine of the ten transgenic lines showed no infection by PVY^O^, and six of the nine showed no infection by PVY^NTN^.	[108]
	CaCV GBNV CMV ChiVMV	*Orthotospovirus* *Cucumovirus* *Potyvirus*	hpRNA	viral silencing suppressors gene	hpRNA	Efficiently controls multiple viruses	[109]
Third-generation antiviral transgenic strategy	ToLCNDV	*Begomovirus*	amiRNA	*AV1*, *AV1* + *AV2*	Ath-miR319a	High tolerance when targeting AV1 + AV2. Moderate tolerance when targeting AV1.	[87]
	TYLCV	*Begomovirus*	amiRNA	*AC1*+*Rep*	Ath-miR159a	Confer resistance to TYLCV.	[110]
	TSWV	*Orthotospovirus*	syn-tasiRNA	*NSm*+*RdRP*	TAS1c	100% of the plants were resistant.	[111]
	TSWV	*Orthotospovirus*	syn-tasiRNA	*RdRP*	TAS1c	Delay of viroid accumulation	[112]
	PhCMoV, ToBRFV	*Alphanucleorhabdovirus* *Tobamovirus*	amiRNA	*L*, *M*, *G*	To-miR6026	Bioinformatic assay showed successful results in controlling both viruses.	[93]
	PVY	*Potyvirus*	amiRNA	*CI*, *NIa*, *NIb*, *CP*	Ath-miR319a	Higher protection when targeting NIb or CP.	[113]
	PVY, PVX, PSTVd	*Potyvirus* *Potexvirus* *Pospiviroid*	*amiRNA*	*P25*, *HC-Pro*, *Brp1*	Ath-miR159a	Resistance against PVX, PVY, and PSTVd coinfection simultaneously, whereas the untransformed controls developed severe symptoms.	[92]

PRSV: papaya ringspot virus; ToLCV: tomato leaf curl virus; ToLCNDV: tomato leaf curl New Delhi virus; ToCMoV: tomato chlorotic mottle virus; CaCV: capsicum chlorosis virus; GBNV: groundnut bud necrosis virus; PhCMoV: physostegia chlorotic mottle virus; ToBRFV: tomato brown rugose fruit virus.

## 6. Virus Resistance Breeding Based on the CRISPR/Cas Genome Editing

CRISPR/Cas (clustered regularly interspaced short palindromic repeats/CRISPR-associated) systems have recently emerged as efficient genome editing tools that provide a new breeding strategy for crop breeding against pathogens [114]. CRISPR/Cas has been successful in breeding some crops for pathogen resistance, such as wheat resistant to rust fungi [115], rice resistant to bacterial blight [116], and tomatoes resistant to viruses [114,117].

The CRISPR/Cas system comprises two key components: the guide RNA (gRNA) that complements the target editing sequence and the Cas endonuclease that cleaves the sequences targeted by the gRNA [118]. Cas endonuclease can be divided into two distinct classes (I and II) and six types (I to VI) based on their functional mechanisms [119]. Class I includes types I, III, and IV, which utilize a multi-protein effector complex, while Class II includes types II, V, and VI, which utilize a single effector protein, conferring it a wider adaptability than Class I [120,121]. The CRISPR/Cas system is utilized in two ways in virus resistance breeding: (1) targeting the viral genomic sequence for gene editing by cleaving or mutating the viral genome to inhibit viral replication in the host [41,122,123] and (2) knocking out or mutating the host susceptibility genes involved in virus infection and replication process to reduce the compatible interaction between host and the virus [124].

### 6.1. CRISPR/Cas Genome Editing Targeting DNA Viruses

CRISPR/Cas9, which is currently widely used in gene editing, belongs to Class II and Type II [121]. Since DNA viruses can form dsDNA intermediates during replication, the CRISPR/Cas system can be used to target viral DNA sequences for cleavage or mutation to inhibit viral replication. TYLCV was the first geminivirus to be edited by CRISPR/Cas9 for TYLCV-resistant tobacco breeding [125]. This method has been successfully used on tobacco, *Arabidopsis*, and tomato to generate multi-generational stable resistance against TYLCV, demonstrating the great potential of CRISPR/Cas in anti-geminivirus breeding (Table 4).

For the CRISPR/Cas-mediated engineering of tomato against geminiviruses, the intergenic (IR), CP, and replication (Rep) regions of the geminiviruses were selected as target sites of gRNA (Table 4). The genus *Geminivirus* has a conserved sequence (5′-TAATATAC-3′) in the IR region. Therefore, an IR-gRNA targeting this conserved sequence could be used to develop a broad-spectrum resistant tomato that is resistant to various geminiviruses, including TYLCV, cotton leaf curl kokhran virus (CLCuKoV), and merremia mosaic virus (MeMV). This could significantly reduce the virus accumulation and alleviate disease symptoms in tomatoes [125,126].

### 6.2. CRISPR/Cas Genome Editing Targeting RNA Viruses

The Cas endonucleases of the CRISPR/Cas system targeting plant RNA viruses mainly include FnCas9 (discovered from *Francisella novicida*) belonging to Type II of Class II of the Cas nuclease family [127] and CRISPR/Cas13 (formerly known as C2c2) belonging to Type VI of Class II [128]. Cas13 can be divided into several groups, including Cas13a, Cas13b, Cas13c, etc. [129,130,131]. Cas13a is the first direct homolog of the Cas13 family used to cleave single-stranded RNA (ssRNA) fragments in CRISPR/Cas-mediated gene editing [129]. So far, the CRISPR/FnCas9 and CRISPR/Cas13a systems have been successfully used to edit potatoes, tobacco, rice, sweet potato, and other crops for resistance against various RNA viruses. These viruses include PVY, TMV, CMV, southern rice black-streaked dwarf virus (SRBSDV), rice stripe mosaic virus (RSMV), etc. (Table 4). The gRNA target site in the RNA viruses is similar to that in DNA viruses, which is mainly located in the IR region and some key regions of the RNA virus coding protein (Table 4). Unfortunately, CRISPR/Cas-mediated editing against RNA viruses has not been reported in tomatoes.

**Table 4 ijms-24-15448-t004:** CRISPR/Cas gene editing system used to create virus resistance in tomatoes and other crops.

Target Virus	Genus	Plant	Targeted Genome	CRISPR/Cas	Targeted Region	Efficiency	Reference
TYLCV	*Begomovirus*	Tomato	Viral DNA	CRISPR/Cas9	*CP*, IR	Significant reduction or delayed accumulation of viral DNA compared to the control plants.	[132]
TYLCV	*Begomovirus*	Tomato	Viral DNA	CRISPR/Cas9	*CP*, *Rep*	Low accumulation of the viral DNA genome compared to the control plants.	[114]
TYLCV	*Begomovirus*	Tomato	Viral DNA	CRISPR/Cas9	IR, *CP*, *RCRII*	Reduction or delayed accumulation of viral DNA, abolishing or significantly attenuating symptoms of infection.	[125]
CLCuKoV	*Begomovirus*	Tomato	Viral DNA	CRISPR/Cas9	IR, *CP*, *RCRII*	Significantly limits CLCuKoV and MeMV replication and systemic infection.	[126]
MeMV	*Begomovirus*	Tomato	Viral DNA	CRISPR/Cas9	IR, *CP*, *RCRII*		
CMV	*Cucumovirus*	*Arabidopsis*	Viral RNA	CRISPR/FnCas9	ORF1a, ORF *CP*, 3′UTR	Significantly attenuated infection symptoms and reduced viral RNA accumulation. The resistance was inheritable, and the progenies showed significantly low virus accumulation.	[133]
TMV	*Tobamovirus*	Tobacco	Viral RNA	CRISPR/FnCas9	3′ORFs		
TuMV	*Potyvirus*	Tobacco	Viral RNA	CRISPR/LshCas13a	*HC-Pro*, *CP*	Targeting the HC-Pro rather than those targeting the coat protein (CP) sequence significantly inhibits TuMV-GFP accumulation and systematic movement.	[134]
TuMV	*Potyvirus*	*Arabidopsis*	Viral RNA	CRISPR/LshCas13a	*HC-Pro*, *CP*	Significant inhibition of TuMV-GFP accumulation level and systematic movement in T1 and T2 plants.	[135]
PVY	*Potyvirus*	Potato	Viral RNA	CRISPR/LshCas13a	*P3*, *CI*, *NIb*, *CP*	Specifically resistant to multiple PVY strains while having no effect on unrelated viruses such as PVA or Potato virus S.	[136]
TMV	*Tobamovirus*	Tobacco	Viral RNA	CRISPR/LshCas13a	*RdRp*, *MP*, *CP*	Significant reduction or delayed accumulation of viral RNA compared to the control plants.	[137]
SRBSDV	*Fijivirus*	Rice	Viral RNA	CRISPR/LshCas13a	ORF	Abolishing or significantly attenuating symptoms of infection. T3 transgenic plants we tested showed stable resistance to SRBSDV.	[137]
RSMV	*Cytorhabdovirus*	Rice	Viral RNA	CRISPR/LshCas13a	ORF	Abolishing or significantly attenuating symptoms of infection. T3 transgenic plants we tested showed stable resistance to RSMV.	[137]
SPCSV	*Crinivirus*	Sweet potato	Viral RNA	CRISPR/LwaCas13a CRISPR/13d	*RNase3*	Transgenic plants and their grafted plants showed a significant reduction in virus accumulation and were asymptomatic.	[138]
TYLCV	*Begomovirus*	Tomato	Tomato genome	CRISPR/Cas9	*SlPelo*	Knocking out the bialleles of *SlPelo* proved to suppress systematic infection of TYLCV.	[117]
PVY	*Potyvirus*	Tomato	Tomato genome	CRISPR/Cas9	*eIF4E1*	Significant reduction in susceptibility to the *N* strain (PVY-N) but not to the ordinary strain (PVY-O).	[139]
CMV	*Cucumovirus*	Tomato	Tomato genome	CRISPR/Cas9	*eIF4E1*	Viral aphid transmission from an infected susceptible plant to gene-edited plants was reduced compared with the parental control.	[139]
TEV	*Potyvirus*	Tomato	Tomato genome	CRISPR/Cas9	*eIF4E1*	A combination of mutations in regions I and II of *eIF4E1* associates with resistance to several isolates of potyviruses.	[140]
PVY	*Potyvirus*	Tomato	Tomato genome	CRISPR/Cas9	*eIF4E1*	Differences in silent targets showed differences in resistance levels.	[140]
PepMoV	*Potyvirus*	Tomato	Tomato genome	CRISPR/Cas9	*eIF4E1*	Knocking out *eIF4E1* exhibited a significant reduction accumulation of PepMoV but not TEV.	[141]
PVMV	*Potyvirus*	Tomato	Tomato genome	CRISPR/Cas9	*4E2 (eIF4E2)*	Knocking out *eIF4E2* exhibited resistance to six of the eight PVMV isolates but not to other potyviruses.	[142]
TuMV	*Potyvirus*	Tomato	Tomato genome	CRISPR/Cas9	*eIF(iso)4E*	Homozygous mutations and transgene-free T2 and T3 generation in self-pollinating species showed no differences in dry weights and flowering times with wild-type plants under standard growth conditions.	[143]
ToBRFV	*Tobamovirus*	Tomato	Tomato genome	CRISPR/Cas9	*SlTOM1a-e*	Quadruple-mutant plants did not show detectable ToBRFV CP accumulation or obvious defects in growth or fruit production. The quadruple-mutant plants also showed resistance to three other tobamoviruses.	[31]
ToBRFV	*Tobamovirus*	Tomato	Tomato genome	CRISPR/Cas9	*SlTOM1* *SlTOM3*	*SlTOM1a* and *SlTOM3* are essential for the replication of ToBRFV but not for ToMV and TMV.	[144]

### 6.3. CRISPR/Cas Genome Editing Targeting Host Susceptible Genes

Since plant viruses are highly dependent on the host to complete their replication cycle, the replication, assembly, and movement of viruses in plant cells require interaction with host–plant-specific factors, often referred to as susceptibility genes (*S* genes), for successful infection [145]. Editing these susceptible genes via CRISPR/Cas to break compatible interactions between the virus and host can help develop resistance in the host plants [146].

A well-known *S* gene that confers resistance to potyviruses is the *eukaryotic translation initiation factor 4E* (*eIF4E*). eIF4E is a mRNA cap-binding protein that plays a critical role in initiating mRNA translation and regulating protein synthesis [124]. Studies have shown that eIF4E can interact with the viral protein genome-link (VPg) of PVY to promote the translation, replication, and intercellular movement of the PVY [124,147]. In recent years, there have been many reports of using CRISPR/Cas to knock out or mutate the *eIF4E* homologous gene in tomatoes to obtain tomato lines with complete resistance to potyviruses (Table 4). Tobamoviruses, especially TBoRFV, which emerged recently, are another important group of viruses that threaten tomato production. The *TOBAMOVIRUS MULTIPLICATION1* (*TOM1*) gene encoded by *Arabidopsis* is required for the replication of TMV [148]. When *TOM1* was mutated in *Arabidopsis*, the accumulation of TMV was significantly inhibited [148]. The CRISPR/Cas9-mediated knockout of *TOM1* homologs, including *SlTOM1a-e* and *SlTOM3*, in tomatoes resulted in ToBRFV resistance in tomato plants [31,144].

In addition, *SlPelo*, a TYLCV susceptibility gene in tomatoes [149], has been successfully used for CRISPR/Cas-mediated antiviral breeding [117]. Tomatoes with *SlPelo* knockout showed significant inhibition and limited spread of TYLCV [117]. These results show great potential for CRISPR/Cas antiviral breeding targeting host-susceptible genes, but only if the molecular interactions between viruses and hosts are well-understood.

## 7. Future Prospects

### 7.1. CRISPR/Cas-Mediated Tomato Breeding for Resistance against Orthotospoviruses

Orthotospoviruses cause significant yield and quality reduction in tomatoes [150]; however, there are limited technologies to control these viruses at present. Some progress has been made in the study of TSWV resistance genes, particularly the *Sw-5* gene and its homologs, which have been identified in tomatoes and are widely used in tomato hybrid breeding. However, many TSWV strains have been reported to break the *Sw-5*-mediated resistance in tomatoes worldwide [12,151]. In addition, some orthotospoviruses, such as capsicum ring spot virus (CYRSV), have been reported to break down the *Sw-5*-mediated resistance in tomatoes [152]. These results indicate that it is urgent to develop new methods for tomato resistance breeding against orthotospoviruses.

So far, gene editing-mediated tomato breeding has achieved great success in developing resistance against geminiviruses, tobamoviruses, and potyviruses. However, there are no reports on gene editing-mediated breeding for resistance against orthotospoviruses. The gRNA targeting some critical genes of orthotospoviruses, such as nuclear protein *N* and RNA silencing suppressor *NSs*, may be a good strategy to create tomato lines resistant against orthotospoviruses.

Targeting host-susceptible genes via the gene-editing method is also a viable strategy for tomato breeding against orthotospoviruses. Several host factors interacting with TSWV have been identified, one such factor being the eukaryotic translation elongation factor 1A (eEF1A). eEF1A interacts with the RNP of TSWV, and silencing *eEF1A* via virus-induced gene silencing (VIGS) significantly inhibits TSWV replication in tobacco [153,154]. Ribosomal protein S6 (RPS6) is a host factor that is part of the 40S ribosomal subunit. Silencing *RPS6* showed high resistance to TSWV in tobacco [155]. The suppressor of the G2 allele of skp1 (SGT1) is a co-chaperone that interacts with Hsp70 [156]. Tobacco SGT (NbSGT1) interacts with TSWV NSm to promote intercellular and systemic motility of the virus [157]. Thus, the homologs of these susceptibility factors in tomatoes could be a potential target for gene editing against orthotospoviruses.

### 7.2. Challenges of the CRISPR/Cas-Mediated Antiviral Breeding in Tomato

CRISPR/Cas genome editing technology has great advantages compared with conventional breeding techniques. These advantages include shortening breeding cycles and saving breeding costs. However, at present, CRISPR/Cas mainly relies on the *Agrobacterium*-mediated transgene method to integrate exogenetic DNA segments into the plant genome, making its application controversial and as strictly regulated as that of GM crops. This also restricts the future commercial application of CRISPR/Cas in resistance breeding. Therefore, the development of DNA-free genome editing methods can help to avoid the abovementioned problems. The current DNA-free genome editing techniques include four main methods. (1) Selecting mutant plants without CRISPR/Cas elements from the gene-edited selfed or hybrid progenies. The selection process usually takes a lot of time and effort; therefore, new rapid selection techniques have been developed, such as inserting visible fluorescent markers into CRISPR/Cas vectors to improve selection efficiency [158]. (2) Polyethylene glycol (PEG)-mediated transient expression of CRISPR/Cas system in plant protoplast. The biggest challenge of this method is the difficulty of plant regeneration from protoplasts. Fortunately, gene editing in wild tomatoes has been successful with the PEG-mediated protoplast method [159]. (3) Modifying the plant virus as the vector to deliver the CRISPR/Cas system. So far, various plant viruses have been modified to deliver gRNA, including DNA viruses such as TYLCV, Bean yellow dwarf virus (BeYDV) [160,161], and wheat dwarf virus (WDV) [162]; RNA viruses such as TSWV [163], barley stripe mosaic virus (BSMV) [164], tobacco rattle virus (TRV) [165], and sonchus yellow net rhabdovirus (SYNV) [166] have also been used for gRNA delivery. Plant viruses have many advantages as delivery vectors, including their ease of manipulation, high accumulation levels (including gRNA and repair templates), and systemic movement across host plants, leading to high expression levels of the gRNA in specific tissues, such as flowers, fruits, buds, etc. However, as gRNA carriers, the carrying capacity of plant viruses is greatly limited (typically < 1 kb) [167]. For these reasons, the future development direction of a plant-virus-based CRISPR/Cas delivery system for tomatoes would be to expand the delivery capacity by modifying more potential viruses that can carry large DNA/RNA sequences. (4) Utilizing endogenous mobile mRNA from plants as carriers for gRNA delivery. Some plant endogenous mRNAs can be used for long-distance transport through the phloem. Studies found that a key motif of mRNA, tRNA-like sequence (TLS), is critical for the long-distance transport of mRNA in the phloem [168]. The studies constructed TLS into the Cas9 vector and expressed Cas9/gRNA-TLS construct in the rootstock through transgenic methods. Thus, the gRNA could also be delivered from rootstock to the scion with the assistance of TLS sequences, and the DNA-free genome editing seeds can be selected from the scion plants [169]. This method utilizes the plant endogenous delivery system, making it much more effective than the exogenic delivery system. In addition, this method does not require additional removal of transgenes or regeneration of plants from the transfected protoplasts, meaning that the breeding cycle is greatly shortened and the efficiency of gene editing is improved. Tomato is also a very suitable crop for grafting, and grafting is conducive to the polymerization of multiple resistances in commercial tomato production [169]. However, the current research on mobile mRNA has mainly focused on model plants such as *Arabidopsis* and tobacco, and little is known about the tomato’s mobile mRNA. Therefore, more research needs to be carried out in the future to explore the potential of the mobile mRNA of tomatoes as a gRNA delivery vector.

A potential risk of CRISPR/Cas-mediated antiviral breeding is that targeting the viral genetic sequence could potentially accelerate virus evolution and the breakdown of the host’s resistance. For example, the gene-edited cassava materials targeting the AC2 and AC3 sequences of the African cassava mosaic virus (ACMV) did not show significant resistance compared with the control group. Further sequencing studies found that 33% to 48% of the edited viruses evolved a conserved single-nucleotide mutation at the target recognition site to evade the gRNA recognition and avoid CRISPR/Cas9 cleavage [170]. Therefore, simultaneously targeting multiple sites of the viral genome may be a better way to reduce the risk of viruses evading recognition by gRNA due to single-site mutations [171].

In addition, there is another practical issue that needs to be considered; that is, the possible negative effects of antiviral gene editing on plants need to be evaluated. Balancing the yield, quality, and resistance to disease is a daunting challenge in crop breeding due to the negative relationship among these traits [172,173]. Genome editing in dealing with viruses for virus resistance may also affect other important agricultural traits and agronomic traits of crops, such as yield, biotic and abiotic resistance, etc. [174,175]. Unfortunately, no detailed data on antiviral gene-edited field crops in the field have been reported so far.

### 7.3. Suggestions for Improving Tomato Antiviral Breeding in the Future

We summarized the advantages, challenges, and prospects of three different antiviral breeding strategies, including MAS-based hybrid breeding, RNAi-based transgenic breeding, and CRISPR/Cas-based gene editing, as presented in Table 5.

For future MAS-based hybrid breeding against tomato viruses, the biggest challenge is how to identify novel antiviral genes in tomatoes to address emerging viruses or strains. Based on the arms race pathogen–host model of concerted evolution, we suggest a reinforced integration of ecological, biogeographical, and genetic discipline of tomato and tomato viruses, which will help us identify tomato’s local adaption and source of resistance [176]. Gene regulatory networks via monogenetic methods will help to explain the heritability of antiviral traits [176]. Selection of breeding methods, such as genomic prediction (GP), genome-wide association studies (GWAS), and major QTL mapping, should also be utilized to identify potential new resistance genes. These techniques have already been successful in identifying new virus resistance genes [176,177,178,179]. To overcome the challenges of developing broad-spectrum resistance against multiple viruses, we suggest utilizing the gene pyramiding method through MAS to integrate multiple resistance genes into a single plant in the shortest possible time [79].

For RNAi-based transgenic breeding against tomato viruses, public controversy and strict regulation of policies on GM crops is the biggest challenge. How to improve transgenic technology from the technical level to eliminate public concerns becomes the key focus of future research.

Although CRISPR/Cas-based gene editing has great potential for breeding tomatoes against viruses, several challenges need to be addressed to ensure its application in future antiviral breeding. These include developing DNA-free gene editing technology, improving the efficiency of gene editing and reducing the off-target rate, and developing multiple targets gene editing technology and the *S* gene editing targets.

## 8. Conclusions

Despite many years of agricultural research, there is still a lack of effective chemical control or agronomic management measures to curb viral diseases. The most effective and suitable method is utilizing natural genetic resistance resources or genetic tools to obtain cultivars resistant to viruses. However, identifying the resistance sources/genes for all plant viruses is difficult, and natural sources/genes could be broken by viral mutation. RNAi transgenic research presents a new, useful way to control viral diseases, but the technique is faced with concerns about GM crops, off-target results, and unwanted recombination. A technology that could provide a viable alternative for virus resistance breeding in the future is CRISPR/Cas-mediated gene editing. The CRISPR/Cas gene-editing technology allows for the removal of the exogenous DNA via technical means, thus ensuring DNA-free editing and avoiding the controversy and policy restriction associated with GM crops. However, there is still a need to further explore the virus–plant molecular interactions, develop the endogenous delivery system based on DNA-free genome editing technology, and evaluate how the technology affects other important crop traits such as yield and tolerance to other biotic and abiotic stresses to optimize the efficiency of gene editing for tomatoes.

## Figures and Tables

**Figure 1 ijms-24-15448-f001:**
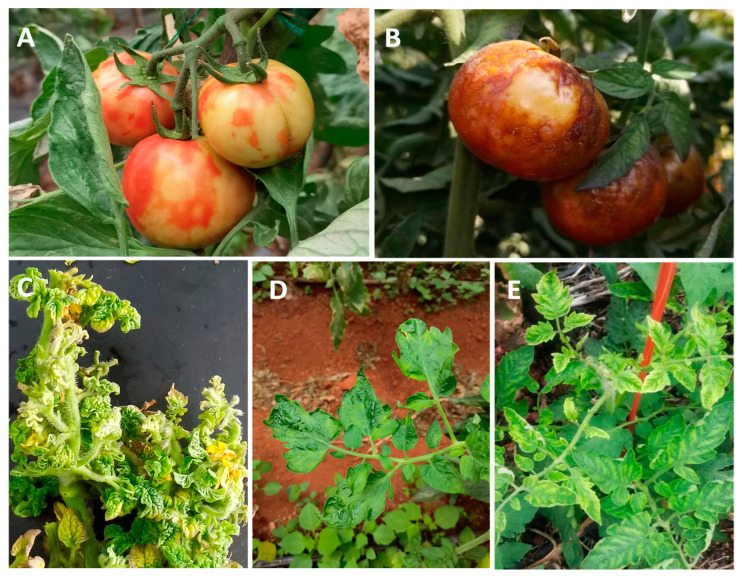
Symptoms of viral diseases in tomatoes. (**A**) Symptoms of TSWV on tomato fruits. (**B**) Symptoms of ToBRFV on tomato fruits. (**C**) Symptoms of TYLCV on tomato leaves. (**D**) Symptoms of ChiVMV on tomato leaves. (**E**) Symptoms of ToCV on tomato leaves.

**Figure 2 ijms-24-15448-f002:**
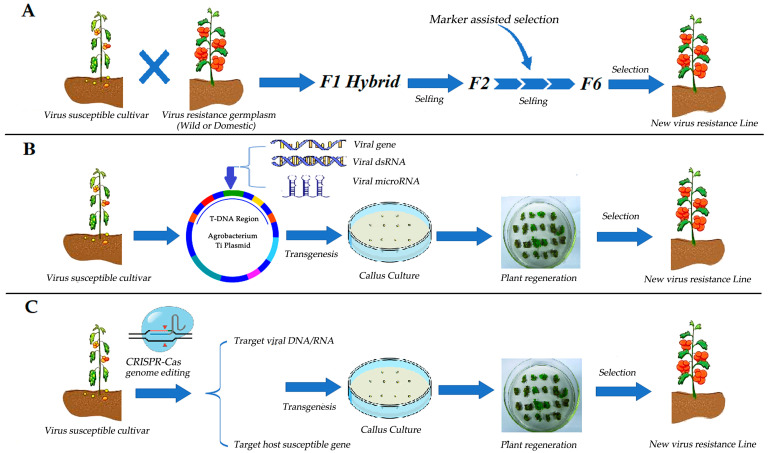
Strategies of tomato resistance breeding against viruses. (**A**) Marker-assisted selection (MAS)-based hybrid breeding. (**B**) RNA interference (RNAi)-based transgenic breeding. (**C**) CRISPR/Cas-based gene editing.

**Table 2 ijms-24-15448-t002:** A brief review of virus resistance gene families in tomatoes.

Resistance Gene Family	Resistance Genes	Source of Resistance Genes	Location on Chromosome	Gene Action	Efficiency	Resistance Mechanism	References
*Ty* gene family (against begomoviruses)	*Ty-1*	*Solanum chilense*	6	Dominant	Broad-spectrum begomoviruses resistance	*Ty-1* encodes an RNA-dependent RNA polymerase (RDR) involved in the RNA silencing pathway, increasing antiviral RNAi responses and the viral genome’s cytosine methylation.	[8,42,44,45,46,47,51,52,53,54]
	*Ty-2*	*S. habrochaites*	11 (Long arm)	Dominant	TYLCV resistance	*Ty-2* encodes a nucleotide-binding leucine-rich repeat (NLR) protein. The Ty-2 could recognize TYLCV Rep/C1 protein and induce hypersensitive responses (HR) in host plant.	
	*Ty-3*	*S. chilense*	6	Dominant	Complementary resistance	*Ty-3* encodes an RNA-dependent RNA polymerase (RDR) involved in the RNA silencing pathway.	
	*Ty-4*	*S. chilense*	3	Dominant	Increase virus resistance in combination with *Ty-3*	Not reported.	
	*Ty-5*	*Tyking*	4	Recessive	Broad-spectrum resistance	Encodes messenger RNA (mRNA) surveillance factor *Pelota*. Silencing of *Pelota* in a susceptible line rendered the transgenic plants highly resistant.	
	*Ty-6*	*S. chilense*	10	Dominant	Complements the resistance conferred by *Ty-3* and *Ty-5*	Not reported.	
*Sw* gene family (against orthotospoviruses)	*Sw-1a*	*Lycopersicum pimpinellifolium*	Not reported	Dominant	Some degree of resistance to specific TSWV	Not reported.	[10,50,52,53,54]
	*Sw-1b*	*L. pimpinellifolium*	Not reported	Dominant	Some degree of resistance to specific TSWV	Not reported.	
	*Sw-2*	*L. pimpinellifolium*	Not reported	Recessive	Some degree of resistance to specific TSWV	Not reported.	
	*Sw-3*	*L. pimpinellifolium*	Not reported	Recessive	Some degree of resistance to specific TSWV	Not reported.	
	*Sw-4*	*L. pimpinellifolium*	Not reported	Recessive	Some degree of resistance to specific TSWV	Not reported.	
	*Sw-5*	*S. peruvianum*	9	Dominant	High level of resistance to a wide range of TSWV	*Sw-5* belongs to nucleotide-binding leucine-rich repeat (NB-LRR) type *R* gene. *Sw-5* confers resistance by recognizing a 21-amino-acid peptide region of the viral movement protein NSm, triggering immunity response.	
	*Sw-6*	*L. pimpinellifolium*	Not reported	Incompletely Dominant	Some degree of resistance to specific TSWV	Not reported	
	*Sw-7*	*L. chilense*	12	Dominant	Resistance to a wide range of TSWV	Involved in pathogenesis-related (PR) proteins PR1 and PR5-related resistance process.	
*Tm* gene family (against tobamoviruses)	*Tm-1*	*S. habrochaites*	2	Incompletely Dominant	TMV partial resistance	*Tm-1* encodes a protein that binds ToMV replication proteins and inhibits the RNA-dependent RNA replication of ToMV.	[14,55,56,57,58,59,60]
	*Tm-2*	*S. peruvianum*	9	Dominant	TMV partial resistance	*Tm-2* belongs to nucleotide-binding leucine-rich repeat (NB-LRR) type *R* gene, which can recognize movement proteins (MPs) of TMV and ToMV and activate a resistance response.	
	*Tm-2^2^*	*S. peruvianum*	9	Dominant	Confers a more effective TMV resistance	*Tm-2^2^* belongs to nucleotide-binding leucine-rich repeat (NB-LRR) type *R* gene, which can recognize movement proteins (MPs) of TMV and ToMV and activate a resistance response.	
*Pot-1* gene (against potyviruses)	*Pot-1*	*L. hirsutum*	3	Recessive	PVY resistance	Tomato *Pot-1* is the orthologue of the pepper *pvr2-eIF4E* gene, encoding the plant-susceptible *eIF4E1* translation initiation factor protein. Duplicate recessive *Pot-1* genes interrupt the interaction of the potyviruses VPg protein with the eIF4E1, suppressing virus replication.	[61,62,63]

**Table 5 ijms-24-15448-t005:** Suggestions for improving tomato antiviral breeding in the future.

Methods	Characteristics	Challenges	Suggestions and Future Prospects
MAS-based hybrid breeding	i. Relay on conventional breeding method. ii. Widely used in commercial breeding.	i. Long breeding cycles compared with genetic engineering breeding. ii. Lack of natural resistance genes. iii. Single-gene-mediated resistance is easily broken down by viruses. iv. Difficulty in developing broad-spectrum resistance against multiple viruses. v. Needs appropriate and reliable DNA markers.	i. Reinforce integration of ecological, biogeographical, and genetic discipline of tomato and tomato viruses, which will help to identify tomato local adaption and source of resistance against emerging virus species or strains. ii. Utilizing selection and breeding methods, such as genomic prediction (GP), genome-wide association studies (GWAS), and major QTL mapping to identify potential new resistance genes. iii. Finding closed, reliable, and effective new molecular markers for virus resistance genes. iv. Using the gene pyramiding method to polymerize multiple resistances in the shortest possible time.
RNAi-based transgenic breeding	i. Genetic engineering method. ii. Rapidly and efficiently creates broad-spectrum resistance against multiple viruses.	i. Public controversy and strict regulation of policies on GM crops. ii. Homology-dependent gene silencing and unwanted recombination, non-target effects and off-target effects.	i. The third-generation antiviral transgenic technology (amiRNAs and syn-tasiRNAs) has improvements, including more specialized in-targets, to overcome the unwanted recombination and off-target effect. ii. Exogenous application of RNAi. research and development.
CRISPR/Cas-based gene editing	i. Genetic engineering method. ii. Relies on Agrobacterium-mediated transgene technology. iii. Shows great potential in antiviral breeding.	i. Controversial and strictly regulated like GM crops. ii. The technology needs to be improved to meet the needs of commercial breeding. iii. Has the potential risk of accelerating virus mutation and evolution. iv. Inefficient targeting of the *S* gene for editing.	i. Developing DNA-free gene editing technology, including DNA-free selection techniques, and a PEG/virus/endogenous mRNA mediated delivery system. ii. Improving the efficiency of gene editing and reducing the off-target rate by utilizing endogenous promoter and optimized delivery system. iii. Developing gene editing techniques targeting multiple viruses simultaneously or targeting multiple sites of a virus simultaneously. iv. Developing the *S* gene editing targets based on the molecular interaction mechanism between viruses and hosts.

## Data Availability

Not applicable.

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
