# Peer review of "Advances and Prospects of Virus-Resistant Breeding in Tomatoes"

_ijms, 2023, doi:10.3390/ijms242015448_

Round 1
Reviewer 1 Report
The review by Shahriari et al. advances our understating of virus resistance in tomato. The review is well written and condensed, as well as technically appropriate. However, before being able to recommend acceptance, I invite authors to address the following amendments.
First, the abstract (L18) and the end of the introduction section (L48) should properly close with an explicit research questions for the review, hypotheses and expected results (so far only a main goal is described). For instance, do authors hypothesize a polygenic or a Mendelian/biomodal segregation for virus tolerance in tomato? Are traits correlated with virus resistance expected to exhibit antagonistic pleiotropy or conditional neutrality when comparing them in various environments? The literature review could offer a perspective of these results.
Second, methodological speaking authors should comment more clearly how they handled literature compilation and articles selection within the review. After all, a review must have a repeatable methodology as any scientific study. Please implement and report Prisma methodology for this purpose (see here: http://prisma-statement.org/PRISMAStatement/?AspxAutoDetectCookieSupport=1)
Third, concerning the results and tables are very well edited and insightful. Still, authors should include a more comprehensive and replicable literature compilation.
Fourth, when virus resistance is measured and observed in each study is a key issue to consider. Was the virus resistance observed during the flowering phenological phase? What about terminal stress? Please expand by adding a new column within the tables within this information, especially for readers coming more from a breeding side.
Fifth, the discussion, although perceptive, should embrace broader reflections on whether virus resistance in tomato could be understood as a plastic or alternative a genetic adaptive strategy to cope with virus incidence. Besides, since biotic and abiotic stresses are usually concurrent, please also comment on potential gene functional correlates or trade-offs of virus resistance with drought and flooding tolerance, which are stresses typically correlated with virus incidence in the face of hypoxia and osmotic imbalance. Any insights in the reported tomato studies?
Sixth, although the paper provides good evidence into the genetic mechanisms for virus resistance in tomato, a major question that authors should prospect in their discussion is how tomato improvement for virus resistance may unlock and effectively utilize the identified trait variation (see as guidance Genes 2021 12:783). Gene editing (already well-described in L235), recurrent backcrossing and inter-specific schemes may offer an avenue in this regard that authors should acknowledge (consider referring to Plants 2021 10:2022). Please envision any other recommendation (some ideas in the next paragraph) in L364 as a perspective section.
Last but not least, the review is so far lacking a very brief closing paragraph (L363) that describes the major caveats/limitations of the compiled works. It is never beyond the scope of any research to explicitly acknowledge the caveats of the state of the art, especially when dealing with highly variable trait profiles in a complex trait with strong GxE effects and plasticity such as virus resistance. For instance, trait variation may exhibit antagonistic pleiotropy or conditional neutrality when comparing their presence and their effects (direction and magnitude) in contrasting environmental treatments, something to be tested in more profound trials beyond a single location). Did the reviewed studies address these matters? Please clarify or recommend as future studies in the short perspectives section, just before the conclusions (L364). Specifically, what other tomato accessions, traits and combine stresses must future studies target? How could the identified segregant candidate genotypes by other studies (something authors should list in more detail as a main new table within the results section inspired in the reviewed studies) source tomato pre-breeding and future improvement programs?
Author Response
- First, the abstract (L18) and the end of the introduction section (L48) should properly close with an explicit research question for the review, hypotheses and expected results (so far only a main goal is described). For instance, do authors hypothesize a polygenic or a Mendelian/biomodal segregation for virus tolerance in tomato? Are traits correlated with virus resistance expected to exhibit antagonistic pleiotropy or conditional neutrality when comparing them in various environments? The literature review could offer a perspective of these results.
Response: According to the suggestion, we have proposed explicit research questions and breeding objective in abstract (L18-28) and the end of the introduction section (L56-63). Tomato virus resistance genes shows qualitative inheritance (monogenic or oligogenic). We add a column to table 2 as gene action. They almost are single qualitative locus with complete dominant and biomodal segregation, in the challenges in tomato hybrid breeding against viruses we described in line232-253.
- Second, methodological speaking authors should comment more clearly how they handled literature compilation and articles selection within the review. After all, a review must have a repeatable methodology as any scientific study. Please implement and report Prisma methodology for this purpose (see here: http://prisma-statement.org/PRISMAStatement/?AspxAutoDetectCookieSupport=1)
Response: We have conducted Prisma checklist, see supplements materials.
- Third, concerning the results and tables are very well edited and insightful. Still, authors should include a more comprehensive and replicable literature compilation.
Response: New update references and informative results were added to the results and tables2, 3
- Fourth, when virus resistance is measured and observed in each study is a key issue to consider. Was the virus resistance observed during the flowering phenological phase? What about terminal stress? Please expand by adding a new column within the tables within this information, especially for readers coming more from a breeding side.
Response: New informative columns including gene action and resistance mechanism were added to the table 2 and more sources were mentioned in the table. According to the gene-to-gene theory, most of the virus resistance genes found at present are R gene, and the ultimate stress in breeding comes from virus breakdown resistance genes by mutation, recombination, mixed-infection.
- Fifth, the discussion, although perceptive, should embrace broader reflections on whether virus resistance in tomato could be understood as a plastic or alternative a genetic adaptive strategy to cope with virus incidence. Besides, since biotic and abiotic stresses are usually concurrent, please also comment on potential gene functional correlates or trade-offs of virus resistance with drought and flooding tolerance, which are stresses typically correlated with virus incidence in the face of hypoxia and osmotic imbalance. Any insights in the reported tomato studies?
Response: Abiotic stresses may affect the life cycle of viruses as well as the interactions between host susceptibility factors and viruses. Conversely, viruses can influence the plant response to abiotic stresses. This matter is considered as research prospect.
Mishra, R., Shteinberg, M., Shkolnik, D., Anfoka, G., Czosnek, H., &Gorovits, R. (2022). Interplay between abiotic (drought) and biotic (virus) stresses in tomato plants. Molecular Plant Pathology, 23(4), 475-488. https://doi.org/10.1111/mpp.13172
Sixth, although the paper provides good evidence into the genetic mechanisms for virus resistance in tomato, a major question that authors should prospect in their discussion is how tomato improvement for virus resistance may unlock and effectively utilize the identified trait variation (see as guidance Genes 2021 12:783). Gene editing (already well-described in L235), recurrent backcrossing and inter-specific schemes may offer an avenue in this regard that authors should acknowledge (consider referring to Plants 2021 10:2022). Please envision any other recommendation (some ideas in the next paragraph) in L364 as a perspective section.
Response: We have added a summary in 6.3, supplementing some of our suggestions for future tomato antiviral breeding (Table4).
Last but not least, the review is so far lacking a very brief closing paragraph (L363) that describes the major caveats/limitations of the compiled works. It is never beyond the scope of any research to explicitly acknowledge the caveats of the state of the art, especially when dealing with highly variable trait profiles in a complex trait with strong GxE effects and plasticity such as virus resistance. For instance, trait variation may exhibit antagonistic pleiotropy or conditional neutrality when comparing their presence and their effects (direction and magnitude) in contrasting environmental treatments, something to be tested in more profound trials beyond a single location). Did the reviewed studies address these matters? Please clarify or recommend as future studies in the short perspectives section, just before the conclusions (L364). Specifically, what other tomato accessions, traits and combine stresses must future studies target? How could the identified segregant candidate genotypes by other studies (something authors should list in more detail as a main new table within the results section inspired in the reviewed studies) source tomato pre-breeding and future improvement programs?
Response: Methods limitation and prospects was added in several new sentences and paragraph in each section. In section 6.3, we summarized the challenges and prospects of the three major breeding technology strategies.

Reviewer 2 Report
Well organized manuscripti, soundly explaining current state of art. The review is summarizing attempts being done accross different virus types goving a good overview for the readers. Aivalable reviews do not summarize the information similar way. Some very minor changes are recommended. Perhaps authors could mention perpectives of RETRON´s.
Some suggestions are inserted in the manuscript. Quality of english correnponds to standards of MDPI editing service.

Usually I m not very happy with language editing - similar to available suggestions of "Word" suggestions.
Even the advanced editing do not provid better "soundness"
Author Response
Response: According to the suggestion, we separated the viruses calssification as a new section. and other marks have also been revised.
Reviewer 3 Report
The review makes and compilation on the literature of the stretgies to cope with viral diseases in tomato. Althoug the information is useful for the community and the review has some interest, needs major improvement before being suitable for publication.
Major points:
The review is arid. There is too much text and very discursive, so is not easy to follow, specially for a broad audience, and IJMS is a general journal not a virology journal, so you mus make the text comprehensible for scietifics not specialist in the topic. I recommend:
Including one or two figures summarizing all the strategies used so far in a braphic manner.
Include and additonal table summarizing all the data included in future prospects.
Lines 171-178: plesea rewritte, as there are several mistakes.
First: Authors say (lines 175-176) that "is far safer than the first generation". GMO are safe, otherways, would have never been approved, nor marketed, and there has never been a proble derived from the use of GMO, so please clarify this.
Second: Authors suggest that including a synthetic RNA which does not express a protein would not be considered a GMO. This is not what most regulations on GMO crops say. Once you include foreing DNA in the genome (synthetic or from other organism) by means of genetic engineering ypu have a GMO crop, and is not cosidered whether this DNA is transcribed to RNA and then translated into a protein or no.
The manuscript needs also thorough revision.
Lines 55-63: Different letter size and parts of the text are in italics.
Lines 112-113: separate "by single".
In other parts of the manuscript appeears some underlining that is out of context. Please eliminate it.
Author Response
The review is arid. There is too much text and very discursive, so is not easy to follow, specially for a broad audience, and IJMS is a general journal not a virology journal, so you mus make the text comprehensible for scietifics not specialist in the topic. I recommend:
- Including one or two figures summarizing all the strategies used so far in a braphic manner. Include and additional table summarizing all the data included in future prospects.
Response: According to the suggestion, we added a figure(Figure 2) summarized the three main breeding strategies. In addition, the prospect of three breeding strategies is also given in the section 6.3.
- Lines 171-178: plesea rewritte, as there are several mistakes.First: Authors say (lines 175-176) that "is far safer than the first generation". GMO are safe, otherways, would have never been approved, nor marketed, and there has never been a proble derived from the use of GMO, so please clarify this.
Response: We have corrected it and rewritte this section.
Second: Authors suggest that including a synthetic RNA which does not express a protein would not be considered a GMO. This is not what most regulations on GMO crops say. Once you include foreing DNA in the genome (synthetic or from other organism) by means of genetic engineering ypu have a GMO crop, and is not cosidered whether this DNA is transcribed to RNA and then translated into a protein or no.
Response: We have corrected it.
The manuscript needs also thorough revision.
Lines 55-63: Different letter size and parts of the text are in italics.
Response: We have corrected it.
Lines 112-113: separate "by single".
Response: We have corrected it.
In other parts of the manuscript appeears some underlining that is out of context. Please eliminate it.
Round 2
Reviewer 1 Report
The authors have substantially improved the manuscript and from my side it would be feasible to recommend acceptance. One last suggestion would be to envision phylo-geographic inferences in the study of tomato genotypes and its natural resistance to virus (or viceversa, the natural resistance of virus to tomato's defences) by referring . Authors may find insightful in this regard the green panel in figure 2 of the review in Plants 2021, 10, 2022 (https://doi.org/10.3390/plants 10102022).
Author Response
Response:Thanks to the reviewer's suggestion. We have carefully read the paper Plants 2021, 10, 2022, which is an insightful and forward-looking paper. We cited some of the viewpoints of this paper about how to discover new resistance resources, please see revised manuscript in section7.3, Line 474-481 for details. In addition, we conducted English language editing through MDPI, and an English editing certificate is provided as supplementary material.
